# Preparation and Corrosion Behavior in Marine Environment of MAO Coatings on Magnesium Alloy

**DOI:** 10.3390/ma13020345

**Published:** 2020-01-12

**Authors:** Yuhong Yao, Wei Yang, Dongjie Liu, Wei Gao, Jian Chen

**Affiliations:** 1School of Materials Science and Chemical Engineering, Xi’an Technological University, Xi’an 710032, China; eifa@sina.com (W.G.); chenjian@xatu.edu.cn (J.C.); 2School of Materials Science and Engineering, Xi’an University of Technology, Xi’an 710048, China; liudongjie@xaut.edu.cn

**Keywords:** magnesium alloy, MAO coating, corrosion behavior, stratification phenomena, marine environments

## Abstract

To improve the corrosion performance of magnesium alloys in the marine environment, the MAO, MAO–Cu_2_CO_3_(OH)_2_·H_2_O and MAO–Cu_2_P_2_O_7_ ceramic coatings were deposited on AZ91D magnesium alloys in basic electrolyte and the discoloration mechanism of the Cu-doped MAO coatings and the corrosion behavior of the three MAO coatings in the artificial seawater solution were investigated by SEM, EDS and XPS. The results indicated that the formation and discoloration mechanism of the brown MAO ceramic coatings were attributable to the formation of Cu_2_O in the coatings. Though the three MAO coatings had a certain protective effect against the corrosion of AZ91D substrate in the artificial seawater, the distinctive stratification phenomenon was found on the MAO–Cu_2_P_2_O_7_ coated sample and the corrosion model of the MAO–Cu_2_P_2_O_7_ coatings in the immersion experiment was established. Therefore, the brown Cu-doped MAO coatings were speculated to significantly reduce the risk of the magnesium parts in marine environments.

## 1. Introduction

The magnesium alloy is the lightest structure metal material, and is considered as the green engineering material in the 21st century [1]. Now it is widely applied to the electronic industry, aerospace industry, and auto industry [2,3]. However, for its poor corrosion resistance, magnesium alloy, particularly as the magnesium alloy parts for outdoor applications, is confronted with great challenges [4,5,6,7]. Many literatures have verified that the surface modification technique, such as chemical conversion coatings, anodic oxidation, micro arc oxidation (MAO), organic coatings, vapor phase processes, etc, is an effective way to change the surface composition and improve its corrosion resistance of magnesium alloy [8,9,10,11,12]. MAO is a newly developed technology for the preparation of ceramic coatings on aluminum, magnesium or titanium alloys to improve the corrosion resistance [10,13,14,15,16]. At present, the synthesis of white MAO coatings on magnesium alloy is a mature and universal technology [16], but it has the same disadvantage as the chemical conversion coating technology, which often causes the formation of the light spots on the surfaces of magnesium diecast components and hardly meets the market demand of 3C (computer, communication and consumer) electronic products. It has been reported that the colored MAO coatings can be formed by adding metal salts in the electrolytes [17,18,19,20,21,22,23,24]. The black MAO coating containing V_2_O_3_ can be obtained on an aluminum alloy surface by adding ammonium metavanadate into the commonly used (NaPO_3_)_6_ (sodium hexametaphosphate) and Na_2_SiO_3_ solution [21] and the black MAO coating with excellent properties can also be prepared in the electrolyte with dichromate addition [22]. Moreover, it has been reported that by adding potassium fluotitanate or sodium stannate into the base electrolyte, a yellow or grey MAO coating can be formed on the surface of Mg alloys, respectively [23,24].

Brown is a very important and common decorative color. Lee et al. [25] has reported that with the addition of 3% and 5% Cu in the base electrolyte, the color of the MAO coating changes from brown to dark brown and the corrosion resistance of the AZ91 alloy is significantly improved after being treated with the micro-arc oxidation process, but the corrosion process and coloring mechanism of this brown MAO coating are still not clear. Furthermore, the magnesium alloy parts with MAO coating for lightweight are sometimes exposed to marine environment and the marine corrosion behavior of the coatings has not been clarified. As a result, the application of magnesium alloy in a marine environment is seriously restricted. In this study, a convenient process to fabricate the MAO coatings with brown color on Mg alloys is introduced by adding alkaline copper carbonate and copper pyrophosphate in the electrolyte and the white and two brown MAO coatings are prepared. Then, the microstructure, formation mechanism and seawater corrosion behaviors of the coatings are investigated in detail. Finally, the seawater corrosion mechanism of the MAO coatings is revealed, which is helpful for the surface protection of magnesium alloy applied in marine environments.

## 2. Materials and Methods

AZ91D magnesium alloys were used as the substrate discs in the size of Ф30 mm × 5 mm and its nominal chemical composition (in wt. %) was Al 8.5–9.5 %, Zn 0.5–0.9 %, Mn 0.17–0.27 %, Cu ≤0.01, Ni ≤0.01, Si ≤0.01, Fe ≤0.004 and Mg balance. Before the micro arc coatings, the specimens were prepared by means of standard metallographic procedure, such as coarse grinding, accurate grinding, polishing with alumina waterproof abrasive paper up to 1200 grit and then ultrasonically degreased in acetone for 10 min followed by rinsing with distilled water.

The MAO coatings were prepared on the specimen surface by using of micro arc oxidation equipment (JHMAO-60, China) with the constant voltage of 420 V, the frequency of 400 Hz, the duty cycle of 10% and the treatment time of 8 min. The base electrolyte solution was composed of 8.0 g/L sodium silicate (Na_2_SiO_3_·9H_2_O), 5.0 g/L potassium hydroxide (KOH), 5.0 g/L potassium fluoride (KF), 1.0 g/L EDTA (C_10_H_16_N_2_O_8_) and 3.5 g/L potassium sodium tartrate (C_4_H_4_O_6_KNa·4H_2_O). The two Cu-doped brown coatings were prepared by respectively adding 2.5 g/L basic copper carbonate (Cu_2_CO_3_(OH)_2_·H_2_O) and 2.5 g/L copper pyrophosphate (Cu2P2O7) into the base solution and the temperature of the electrotype with pH of about 13 was kept below 35 °C during the MAO process.

The thickness of the MAO coatings was measured by using TT240 eddy current thickness meter with an accuracy of 0.1 μm. Six measurements were carried out evenly on the whole sample surface. The surface morphologies and element distribution of the MAO coatings were analyzed by scanning electron microscope (SEM) with Oxford Inca X-Max energy dispersive spectrometry (EDS). X-ray photoelectron spectroscopy (XPS) with Al (mono) Kα irradiation at pass energy of 160 eV (AXIS UTLTRADLD) was used to characterize the chemical bonds of the coatings. The binding energies were referenced to the C 1 s line at 284.6 eV from adventitious carbon. The corrosion behavior of the coated AZ91D magnesium alloy was evaluated by the immersion tests in the artificial seawater, whose composition was shown in Table 1. Before the immersion test, the three MAO-coated specimens were treated with epoxy resin to avoid the effect of defects at the edge of the samples, then immersed in the artificial seawater solution for 14 days and the corrosion morphologies of the samples were observed by SEM.

## 3. Results and Discussions

The MAO–Cu_2_CO_3_(OH)_2_·H_2_O and MAO–Cu_2_P_2_O_7_ coatings on AZ91D magnesium alloys in Figure 1 are prepared with the thicknesses of 8.4 µm, 9.1 µm and 9.6µm, respectively. It reveals that for the more intense micro arc discharge many defects are observed at the edge of the sample in Figure 1b. The microstructures of the three MAO coatings are given in Figure 2. It can be seen from Figure 2 that the surfaces of the three MAO coatings are characterized by lots of micropores with the size range from submicron to several micro scale, and different from the MAO and MAO–Cu_2_P_2_O_7_ coatings, there are some bumps on the MAO–Cu_2_CO_3_(OH)_2_·H_2_O coating surface, which are formed by the spark discharge and gas bubbles throughout the discharge channels during MAO process [26,27]. So it reveals that the addition of Cu_2_CO_3_(OH)_2_·H_2_O into the base electrolyte results in a strongly intense micro arc discharge, which promotes the formation of large molten deposits (Figure 2c,d). The elemental compositions of the three MAO coatings are examined by EDS in Table 2. Carbon (C) is considered to be an impurity from the atmosphere or the electrolyte, F and Na are also presumed to originate from the electrolyte or the AZ91D alloy substrate, P and Si are from the electrolyte and Zn comes from the substrate which indicates that the thickness of the three MAO coatings is very thin. Thus the three MAO coatings are mainly composed of Mg, O, Si and a little amount of C, F, Na species. The presence of Si and O reveals that the components of the electrolyte have intensively incorporated into the micro arc oxidation reactions to from the ceramic coatings [16]. Moreover, by addition of Cu_2_CO_3_(OH)_2_·H_2_O or Cu_2_P_2_O_7_ into the electrolyte, a very small amount of Cu or Cu, P has been respectively doped into the MAO–Cu_2_CO_3_(OH)_2_·H_2_O and MAO–Cu_2_P_2_O_7_ coatings to make the color of the coatings change from white to brown. The discoloration mechanism of the MAO coatings will be further discussed by using SEM + EDS and XPS.

The surface morphologies of the MAO–Cu_2_CO_3_(OH)_2_·H_2_O coatings formed at different oxidation times are shown in Figure 3 and the corresponding EDS analysis results are listed in Table 3. As shown in Figure 3a and Table 3, when the oxidation time is about 70 s, the surface morphology of the AZ91D substrate is inhomogeneous at the moment of starting arc with two distinct regions: region I with a damaged area caused by electrical break-down involving amount of O, F, Si and Cu elements from the electrolyte and Mg and Zn alloying species from the substrate, and region II with a smooth surface morphology and the scratches in the substrate including lower content of O, Si, Cu and much higher content of Mg than those in region I, which indicates that the electrical breakdown phenomenon does not occur in region II. With the prolongation of oxidation time, the surface of the samples presents a typical porous feature and the pore size of the MAO coatings increases with the oxidation time in Figure 3b–d; a similar micro arc process and the mechanism of the pore initiation and the pore development are reported by some literatures [10,21]. However, it is worth noting that the concentrations of O, F, Si, Cu, Mg and Zn elements are similar in the MAO–Cu_2_CO_3_(OH)_2_·H_2_O coatings, the color of the coatings gradually becomes deeper with oxidation time, so it is very meaningful to analyze the discoloration mechanism of the coatings.

The chemical states of Cu, Mg and Si are investigated by using XPS in Figure 4. From the wide spectra demonstrated in Figure 4a, Mg, O, Si and F elements are all found in the three MAO coatings and Cu is detected in the MAO coating with MAO–Cu_2_CO_3_(OH)_2_·H_2_O or Cu_2_P_2_O_7_ addition by XPS. The Cu2p3/2 spectrum consists of two peaks at the binding energies of 932 eV associated with Cu_2_O and 939.7 eV corresponding to CuF_2_ in the brown MAO coating with Cu_2_CO_3_(OH)_2_·H_2_O addition in Figure 4b. Figure 4c,d illustrate that the Mg2p peak at the binding energy of 47.8 eV and the Si2p peak of 100.25 eV are individually assigned to MgO and SiO2. So, it can be concluded that the solute ions (such as Cu and Si) from the electrolyte are involved in the growth process of the MAO coatings, and the same results are found in the growth process of the MAO coatings on Ti substrate [28,29]. Therefore, it is the formation of the red Cu_2_O in the MAO coatings with MAO–Cu_2_CO_3_(OH)_2_·H_2_O addition that results in the discoloration of the coatings.

Figure 5 and Figure 6 are the macro and micro surface appearances of the three MAO coatings immersed in the artificial seawater solution for 14 days, respectively. Compared with Figure 1, it can be observed from Figure 5 that the color of the three MAO coatings obviously changes and some corrosion products are formed on the immersed coatings. The corrosion is more serious at the edge of the three MAO samples due to the defects caused by micro arc discharge (Figure 1b) or the damage during the embedding. It has been reported that once the corrosion reaction is initiated on the sample through a pit or minute pore, the corrosive medium can come into contact with the substrate to form some corrosion pits [30]. The elemental compositions of the three MAO coatings detected by EDS after the immersion test are listed in Table 4. The three corroded MAO coatings are mainly composed of Mg from the substrate and the artificial seawater and O from the electrolyte during the micro arc discharge, and Si, Cu and F from the electrolytes and Cl, K and Ca elements from the artificial seawater are also found in the corroded MAO coatings, which indicate that the MAO coatings are not completely destroyed. From Figure 6, it is quite clear that the microstructures of the MAO coatings have a significant change before and after the immersion test. The white MAO coating exhibits a relatively uniform surface appearance with a high degree of porosity, some cracks and corrosion products as shown in Figure 6a, which indicates that the AZ91D substrate is still protected by the coatings. The MAO–Cu_2_CO_3_(OH)_2_·H_2_O coatings are relatively rougher and exhibit a stacking structure with limited number of pores in Figure 6b. For the MAO–Cu_2_P_2_O_7_ coatings, besides many micropores and some micro-cracks, there is a sedimentary layer with a lot of cracks in the coatings in Figure 6c and SEM morphology and EDS analysis results of this MAO coating after the immersion test are shown in detail in Figure 7 and Table 5. It can be seen from Figure 7 that the corroded coatings are divided into three regions: region I (the inner layer near the Mg substrate), region II (the middle layer attached to the surface coating) and region III (the top layer). From Table 5, Mg, O, F and Si elements are found in the region I and the atom percentage of these elements is similar to those in the MAO coating before the immersion test. Moreover, Cl and Ca elements from the artificial seawater are not observed, indicating that the MAO ceramic coatings in region I have not been destroyed. Region II is a dense layer attached to the MAO coating, where the contents of Mg, Si and F elements dramatically decrease whereas the contents of O element significantly increase compared with that in region I; both Cl and Ca elements have been detected. It indicates that there is the interaction between the MAO coating and the corrosive medium. In the case of region III, this layer is relatively thick and composed of some loose and porous structure sediments, mainly containing O and Ca elements. It is well known that the deposition of corrosion products can hinder the transfer of the charge and increase the inhibition of corrosion, so it is believed that this thick sediment layer is very helpful to improve the corrosion behavior of the MAO–Cu_2_P_2_O_7_ coatings in artificial seawater.

From Figure 7 and Table 5, it has been learnt that the inner layer of the coatings is not destroyed near the substrate during the corrosion process, a middle dense layer is formed in the corrosion solution and the top thick deposition layer is of porous characteristics. Thus, a schematic diagram of the MAO–Cu_2_P_2_O_7_ coatings during the immersion corrosion in the artificial seawater solution is shown in Figure 8. Due to the eruption and condensation of molten materials caused by micro arc discharge, the MAO coating normally has porous characteristics [31]. As a result, some interconnected micro closed-pores exist inside the MAO ceramic layer in Figure 8a. As shown in Figure 8b, the corrosion medium, especially Cl-, can seep into the interface between the electrolyte and the coatings through the micropores during the corrosion process; these micropores are exposed to the corrosive medium due to the oxygen concentration polarization between the interior holes and the interface. Then the corrosion products accumulate in the micropores and form a dense corrosion product layer on the MAO coating with the prolonging of the immersion corrosion time to prevent further corrosion of the coatings. Finally, the further interaction between the substrate and the artificial seawater has effectively been prevented by the corrosion product layer and the top calcium oxide-like thick porous layer forming on the middle dense layer in the following corrosion process in Figure 8c, which can further prevent the corrosion medium into the substrate to enhance its corrosion performance in the artificial seawater. Thus, it can be concluded that the stratification phenomena have been found on the MAO–Cu_2_P_2_O_7_-coated sample and similar results have also been reported in the literatures [32,33]. However, although there is no Mg detected in the top sedimentary layer, for its loose and porous structure there is maybe a risk of corrosion for AZ91D substrate with further extension of the corrosion process.

## 4. Conclusions

(1) Brown MAO coatings on magnesium alloy can be prepared in the Na_2_SiO_3_ alkaline electrolyte with copper pyrophosphate or copper carbonate as the colorant, and for strongly intense micro arc discharge, some bumps appear on the Cu-doped MAO coatings.

(2) With the increase of reaction time, Cu in the colorant is fully involved in the formation of the MAO–Cu_2_CO_3_(OH)_2_·H_2_O and MAO–Cu_2_P_2_O_7_ coatings on AZ91D magnesium alloys, and the discoloration mechanism of these coatings are attributed to the formation of Cu_2_O in the coatings.

(3) After immersion in seawater for 14 days, the stratification corrosion microstructure with the three layers named the inner layer, the middle dense layer and the top calcium oxide-like layer are formed on the MAO–Cu_2_P_2_O_7_ coated sample, which is helpful to prolong the service life of AZ91D magnesium alloys in artificial seawater.

## Figures and Tables

**Figure 1 materials-13-00345-f001:**
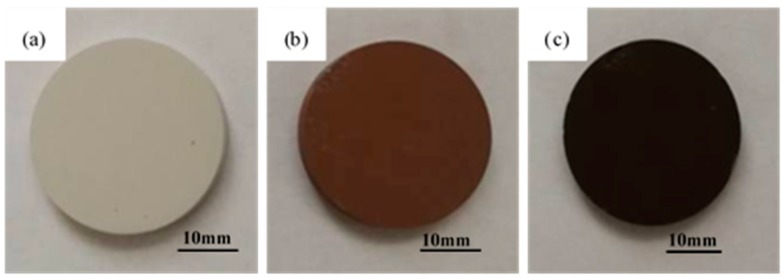
Macrograph of three MAO coatings on AZ91D magnesium alloy, (**a**) MAO, (**b**) MAO–Cu_2_CO_3_(OH)_2_·H_2_O and (**c**) MAO–Cu_2_P_2_O_7_.

**Figure 2 materials-13-00345-f002:**
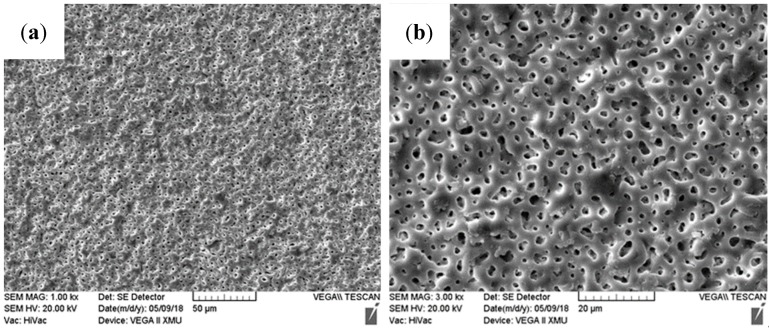
Surface morphologies of three MAO coatings on AZ91D magnesium alloy, (**a**) and (**b**) MAO, (**c**) and (**d**) MAO–Cu_2_CO_3_(OH)_2_·H_2_O and (**e**) and (**f**) MAO–Cu_2_P_2_O_7_.

**Figure 3 materials-13-00345-f003:**
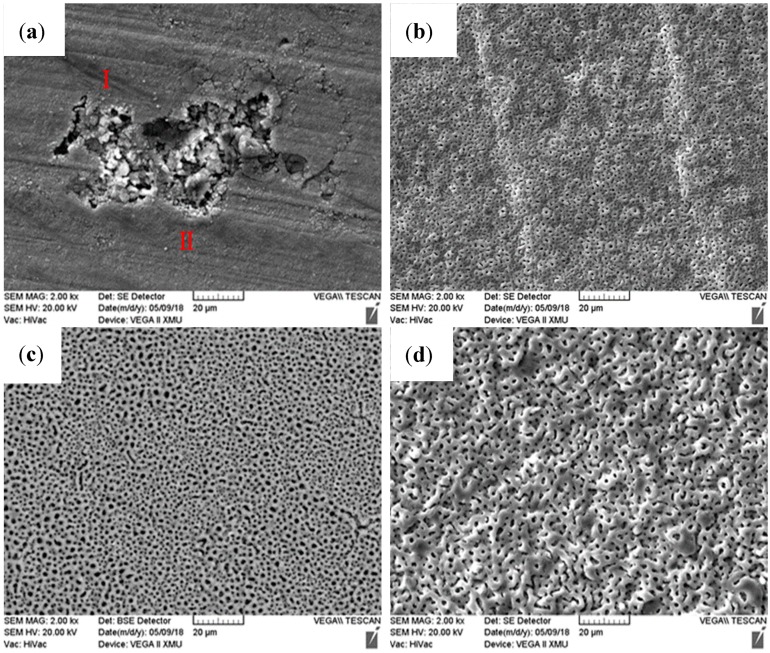
Surface morphologies of MAO–Cu_2_CO_3_(OH)_2_·H_2_O coatings formed with different oxidation time on AZ91D magnesium alloy, (**a**) 70 s, (**b**) 100 s, (**c**) 120 s and (**d**) 180 s.

**Figure 4 materials-13-00345-f004:**
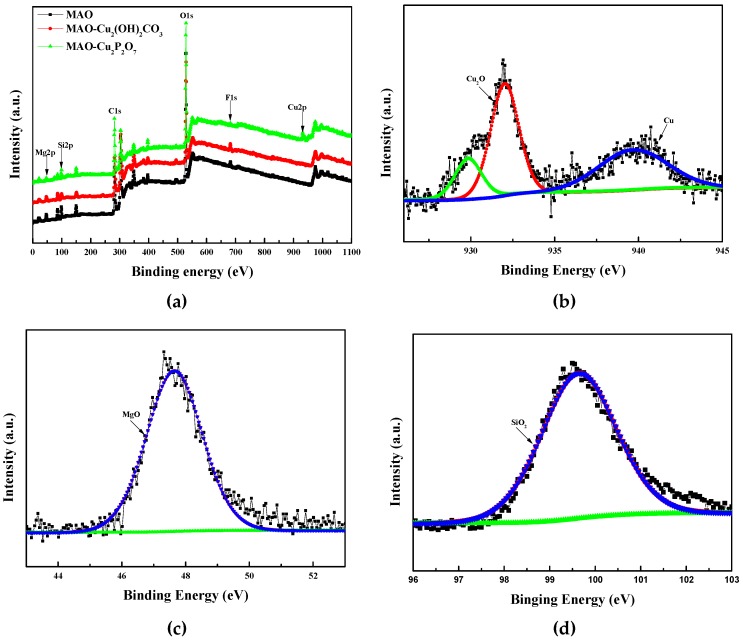
(**a**) XPS survey spectra of three MAO coatings, (**b**) Cu2p, (**c**) Mg2p and (**d**) Si2p typical high-resolution XPS spectrum of MAO–Cu_2_CO_3_(OH)_2_·H_2_O coatings.

**Figure 5 materials-13-00345-f005:**
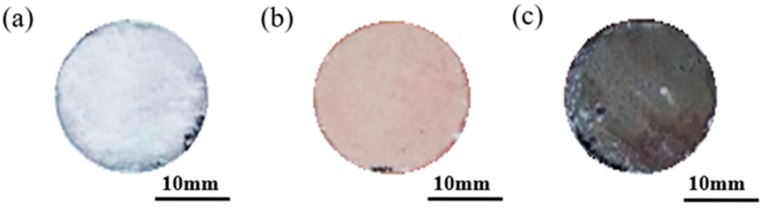
Macrograph of three MAO coatings, (**a**) MAO, (**b**) MAO–Cu_2_CO_3_(OH)_2_·H_2_O and (**c**) MAO–Cu_2_P_2_O_7_, on AZ91D magnesium alloy after 14 days immersion test in the artificial seawater.

**Figure 6 materials-13-00345-f006:**
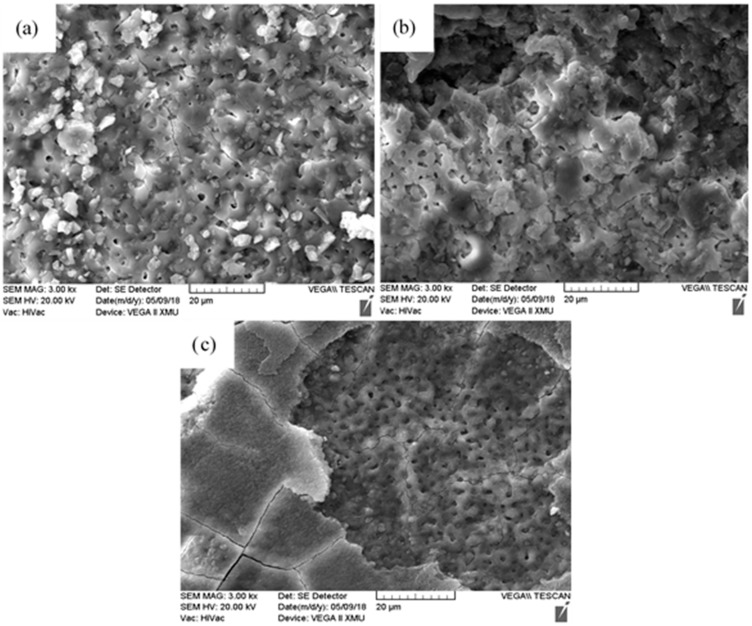
Micro morphologies of three MAO coatings by SEM after immersion test in the artificial seawater for 14 days, (**a**) MAO, (**b**) MAO–Cu_2_CO_3_(OH)_2_·H_2_O and (**c**) MAO–Cu_2_P_2_O_7_.

**Figure 7 materials-13-00345-f007:**
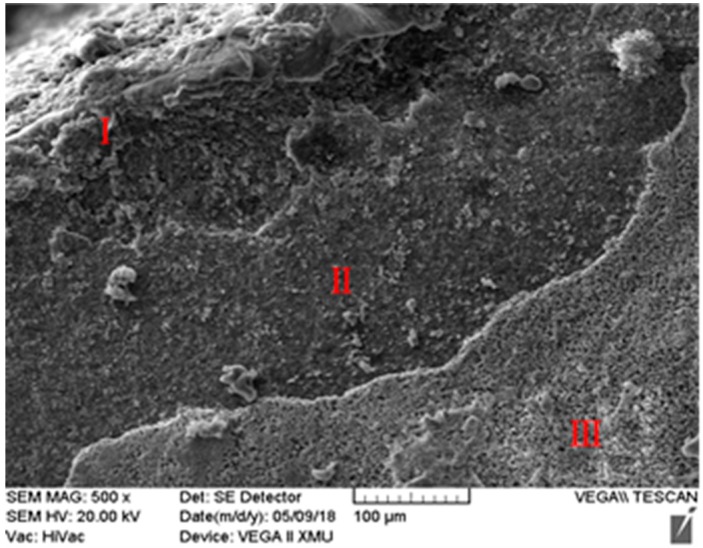
SEM surface morphologies and EDS results of MAO–Cu_2_P_2_O_7_ coating after immersion test.

**Figure 8 materials-13-00345-f008:**
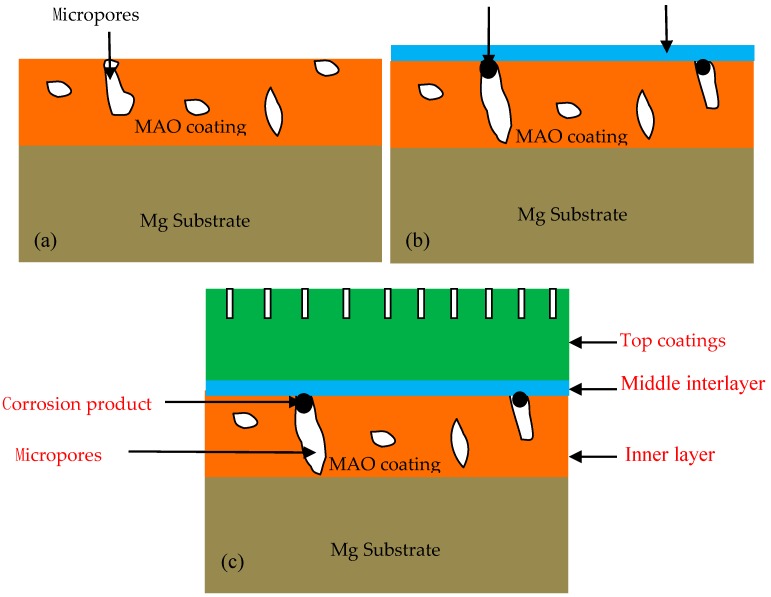
Schematic diagrams of the immersion corrosion of MAO–Cu_2_P_2_O_7_ coating in the artificial seawater. (**a**) Morphology of MAO coating before corrosion, (**b**) Morphology of MAO coating at the initial stage of corrosion, (**c**) Morphology of MAO coating at the latter stage of corrosion.

**Table 1 materials-13-00345-t001:** Composition of artificial seawater in immersion test.

Chemical Reagents	Concentration (g/L)
NaCl	24.53
MgCl_2_·6H_2_O	11.11
Na_2_SO_4_	4.09
CaCl_2_	1.16
KCl	0.70
NaHCO_3_	0.20
KBr	0.10

**Table 2 materials-13-00345-t002:** EDS results of the MAO, MAO–Cu_2_CO_3_(OH)_2_·H_2_O and MAO–Cu_2_P_2_O_7_ coatings on AZ91D magnesium alloy (at. %).

Coatings	C	O	F	Na	Mg	Si	Zn	Cu	P
MAO	1.84	49.64	3.63	0.46	33.98	9.78	0.67	-	-
MAO–Cu_2_CO_3_(OH)_2_·H_2_O	2.95	48.18	3.76	0.86	35.28	8.05	0.39	0.53	-
MAO–Cu_2_P_2_O_7_	2.35	49.42	4.45	0.21	32.42	7.27	1.17	0.74	2.81

**Table 3 materials-13-00345-t003:** EDS results of MAO–Cu_2_CO_3_(OH)_2_·H_2_O coatings formed with different oxidation time on AZ91D magnesium alloy (at. %).

Oxidation Time	C	O	F	Mg	Si	Cu	Zn
70 s—I	2.92	45.97	4.45	36.30	7.73	0.26	2.37
70 s—II	2.05	29.67	2.77	60.81	2.45	-	1.76
100 s	1.04	49.26	5.97	34.99	7.71	0.28	0.75
120 s	1.00	52.41	5.80	32.41	7.65	0.28	0.45
180 s	1.80	50.68	4.55	34.28	8.16	0.12	0.40

**Table 4 materials-13-00345-t004:** EDS results of the three MAO coatings immersed in artificial seawater for 14 days (at. %).

Coatings	O	Mg	Zn	Cu	Si	F	Cl	K	Ca
**MAO**	48.03	29.17	1.66	-	5.72	7.05	0.35	0.26	0.81
**MAO–Cu_2_CO_3_(OH)_2_·H_2_O**	35.25	38.13	2.02	-	4.07	5.92	0.41	-	-
**MAO–Cu_2_P_2_O_7_**	57.89	27.09	0.82	1.04	6.30	-	0.59	-	0.55

**Table 5 materials-13-00345-t005:** EDS results of the three layers of the MAO–Cu_2_P_2_O_7_ coatings immersed in artificial seawater for 14 days (at. %).

Regions	O	Mg	Si	F	Cl	Ca
**Region I**	47.08	30.15	5.72	7.05	-	-
**Region II**	60.47	25.42	2.37	-	0.73	2.57
**Region III**	71.34	0.61	-	-	-	22.09

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
