# Peer review of "Preparation and Corrosion Behavior in Marine Environment of MAO Coatings on Magnesium Alloy"

_materials, 2020, doi:10.3390/ma13020345_

Round 1
Reviewer 1 Report
Correct research work with many interesting new findings, however, its presentation should be improved. Just a few examples:
In line 27: The Magnesium... do not use capital letter for Magnesium
In line 39: What is 3C ?
In line 41. ... can be was obtained...Please correct the grammer!
In line 42: What is (NaPO3)6 ?
In lines 65&66: Not the actual chemical composition is given for the used alloy !
In line 87: instead of "simulated" probably "artificial seawater" would be better
In line 114: Instead of "C is considered..." please start the sentence with "Carbon (C) is...
In the Tables 2 & 3 & 4 & 5 is it justifiable presenting concentrations for O, etc. with 4 valuable digits ??? Plus, the figures are in wt% OR! in at.% OR ???
In lines 196 & 208: "...with some literature...- Do you want to refer to data or information found in some literature sources ?
At the end of Chaper 3 the explanation of the CORROSION STORY about the ..."...sedimentation...and stratification phenomena..." was not really easily understandable for me. THIS PART SHOULD BE IMPROVED, please.
Also in Figure 8 please help the readers understand e.g. the meaning of the black spots, etc., i.e. please correct the drawing with more explanations.
In line 369: instead of "basic" electrolyte, probably using the term "alkaline" would be better.
In line 376: You should probably write just as "After immersion in artificial seawater for 14 days..."
In line 378: "on the" is repeated twice - please correct!
In the References: please chck the lines399, 435, 453, 464,...
Please NOTE that the above mentioned comments are ONLY some examples!
Using "BIG black dots" before the H2O chemical formulas is also not nice.
Reviewer 2 Report
The Authors reported the fabrication of MAO ceramic coatings AZ91D magnesium alloy to improve its corrosion resistance.
The topic of the manuscript is within the scope of the jounral and the obtained results can be interesting to the readers
However, several issues should be addressed before the manuscript is publishable.
The reviewer's suggestions/questions are as follows:
- the novelty aspect of the manuscript should be emphasized at the end of the introduction section
- what was the reason beyond choosing the electrolyte of such composition? Was any preliminary optimization of the MAO conditions performed?
- do you expect any effect of applied voltage on the composition, thikcness, color, and properties of the obtained films? If so, why such voltage value was chosen?
- was the crystallinity of the obtained layers studied?
- why the authors considered that the presence of Zn shown by EDS results is attributed to the substrate and no Zn is present whithin the film? Do you have any proof for that?
- it would be beneficial to perform EDS elemental mapping to study the uniformity of distribution of particular elements within the film, especially near cracks.
- language revision of the manuscript is highly recommended.
Reviewer 3 Report
Interesting paper, thanks for sharing your results.
However, results shown need to be discussed more widely.
Fig. 5 is quite badly understandable when comparing to fig. 1.
Were the samples (shown in fig. 1) embedded? which might explain the sourrounding part in fig. 5. Examples shown in fig. 5 (mainly a) and b)) point to corrosion initiation at the edge, pointing to some crevice effects. Perhaps, during the embedding some coating damage was made. This fact needs to be discussed in more detail.
In order to understand and prove the scheme shown in fig. 8 metallographic cross sections or slope cuttings are necessary.
14 days of exposure are definitely not sufficient to conclude a suitability of a coating.
May be some electrochemical tests, like Electrochemical Impedance Spectroscopy might be supportiv to describe change of layer structure during exposure.
Regarding language: It is quite understandable so far. In conclusion 3) "immersion" should be deleted.
Round 2
Reviewer 2 Report
The manuscript can be published in the present form, since the Authors responded to all questions of the Reviewer. Nevertheless, I would recommend that next time the Authors should emphasize more on improving their manuscript according to Reviewer's suggestions (even by additional comments in the text) rather than on direct replies to the Reviewer. The role of such kind of suggestions is to improve the quality of the manuscript rather than raising the Reviewer's knowledge.
Reviewer 3 Report
Thanks for implementing my comments and answering my questions.
Basing on the response and the revised paper I have only one revision request:
Conclusions 3) Please exchange the big dot by the small one, as you did in the rest of the paper.